# Vitamin E Ameliorates Impaired Ovarian Development, Oxidative Stress, and Disrupted Lipid Metabolism in *Oreochromis niloticus* Fed with a Diet Containing Olive Oil Instead of Fish Oil

**DOI:** 10.3390/antiox12081524

**Published:** 2023-07-29

**Authors:** Yifan Tao, Yifan Pan, Qingchun Wang, Siqi Lu, Yan Li, Wenting Liu, Tao Zheng, Bei Wang, Jun Qiang, Pao Xu

**Affiliations:** 1Key Laboratory of Freshwater Fisheries and Germplasm Resources Utilization, Ministry of Agriculture and Rural Affairs, Freshwater Fisheries Research Center, Chinese Academy of Fishery Sciences, Wuxi 214081, China; 2Wuxi Fisheries College, Nanjing Agricultural University, Wuxi 214081, China; 3College of Fisheries, Guangdong Ocean University, Zhanjiang 524088, China

**Keywords:** olive oil, vitamin E, oxidative stress, *Oreochromis niloticus*, ovarian development

## Abstract

Aquaculture feed containing olive oil (OO) instead of fish oil (FO) can cause oxidative stress and impair gonad development in fish. We determined the effect of dietary OO-induced oxidative stress on ovarian development, and explored whether vitamin E (VE) could mitigate negative effects. Female Nile tilapia (*Oreochromis niloticus*) were fed for 10 weeks with four diets: 5% OO + 70 mg/kg VE, 5% OO + 200 mg/kg VE, 5% FO + 70 mg/kg VE, or 5% FO + 200 mg/kg VE. Dietary OO reduced the specific growth rate and gonadosomatic index, inhibited superoxide dismutase and catalase, delayed ovarian development, decreased serum sex hormone levels, and reduced ovarian triglyceride and n-3 highly unsaturated fatty acid contents. The transcript levels of genes encoding sex hormone receptors (*erα*, *fshr*, *lhr*) and components of the lipid metabolism pathway (*pparα*, *pparγ*, *hsl*, *accα*, *elovl6*), the *nrf2* signaling pathway (*nrf2*, *keap1*), and the *nf-κb* signaling pathway (*nf-κb*, *tnfα*, *infγ*, *il1β*) differed between the 70VE/OO and 70VE/FO groups. Supplementation with 200 mg/kg VE mitigated the adverse effects of OO by improving antioxidant capacity and alleviating inflammation and abnormal lipid metabolism. This may be because VE is an antioxidant and it can regulate the *nrf2*-*nf-κb* signaling pathway.

## 1. Introduction

Traditionally, fish oil (FO) has been used as an important component of aquatic feed. However, because of the strong demand and insufficient supply, the price of FO has continued to rise [1]. Vegetable oil is regarded as a favorable alternative to FO because of its cost-effectiveness and wide availability [2]. Oleic acid (OA, C18:1n-9) is a prominent fatty acid found in vegetable oils, particularly in olive oil (OO) and rapeseed oil (RO). Studies on large yellow croaker (*Larichthys crocea*) [3] and golden pompano (*Trachinotus ovatus*) [4] have shown that OO or RO can replace FO to some extent without affecting growth performance. Fatty acids are an important component of the cell phospholipid bilayer, and imbalanced fatty acid intake can make this bilayer prone to peroxidation. This can cause damage to cell structure and function, resulting in abnormal secretion of cytokines, increased production of reactive oxygen species (ROS), disordered lipid metabolism, and oxidative stress damage. Research has shown that excessive replacement of FO with OO or RO can lead to abnormal lipid deposition and decreased antioxidant capacity in fish [3,5]. For example, Li et al. [3] showed that total replacement of FO with OO activated the p38 MAPK and JNK pathways. These changes resulted in increased hepatic lipid deposition, an inflammatory response, and oxidative stress damage, which ultimately weakened the growth performance of those fish. However, the underlying mechanisms of oxidative damage and aberrant lipid metabolism triggered by excessive OA in fish have not been thoroughly elucidated.

The type of dietary lipids, especially the composition and proportion of long chain polyunsaturated fatty acids (LC-PUFA) in oil, strongly affect gonad development and the reproductive performance of fish. Fish oil is an excellent lipid source for broodstock of fishes, including pearl gourami (*Trichogaster leeri*) [6] and gilthead seabream (*Sparus aurata*) [7], because of its high LC-PUFA content. Studies have shown that partial replacement of FO with OO or RO can promote the reproductive performance of tongue sole (*Cynoglossus semilaevis*) [8], Atlantic salmon (*Salmo salar*) [9], and sterlet sturgeon (*Acipenser ruthenus*) [10], although excessive replacement of FO was reported to decrease the reproductive performance of sterlet sturgeon [10]. However, little is known about the effects of dietary OO on the gonad development and reproduction of tilapia.

As a fat-soluble vitamin, vitamin E (VE) is involved in many processes in fish, such as growth, breeding, the antioxidant system, and intracellular signal transmission [11]. Firstly, VE functions as a nonenzymatic antioxidant to scavenge free radicals by combining with oxides. It can also inhibit the lipid peroxidation, thus reducing oxidative stress damage in fish [12]. Research has shown that an appropriate dietary VE level can enhance the antioxidant capacity in hybrid grouper (♀ *Epinephelus fuscoguttatus* × ♂ *Epenephelus lanceolatus*) [13] and Caspian trout (*Salmo caspius*) [14]. A deficiency of dietary VE in grass carp (*Ctenopharyngodon idella*) was shown to induce oxidative damage, which is closely associated with reduced activities of superoxide dismutase (SOD), catalase (CAT), and glutathione peroxidase [15]. The addition of dietary VE can also positively affect animal reproduction. For example, dietary VE was shown to promote ovarian development, improve the spawning rate, and increase larval quality in red swordtail (*Xiphophorus helleri*) [16] and Black Sea trout (*Salmo labrax*) [17]. In addition, VE plays a vital role in the regulation of lipid metabolism and a deficiency of VE in the diet has been found to result in abnormal lipid deposition in fish [13,18]. For instance, Liang et al. [13] found that an optimal dietary level of VE alleviated lipid accumulation by upregulating the gene encoding hormone-sensitive triglyceride lipase (HSL) and downregulating the gene-encoding fatty acid synthase (FAS) in the liver of hybrid grouper. Dietary VE has been shown to be beneficial to alleviate oxidative damage caused by metal pollution [19], oxidized fish oil [20] and high-fat feed [21]. However, few studies have explored whether VE can alleviate the stress response to excess dietary OA. 

Tilapia is the main freshwater aquaculture fish in the southern provinces of China. China’s tilapia yield in 2021 exceeded 1,650,000 tons, ranking first in the world [22]. However, in recent years, the tilapia aquaculture industry in China has encountered significant degradation of germplasm resources. This has been manifested as a decline of fertility, slow growth, and more severe diseases [23]. Therefore, it is crucial to assess the potential of nutrients to enhance the ovarian development and reproductive performance of tilapia. Both FO and OO are now widely used as lipid components of tilapia feed. However, prolonged intake of OA-enriched diet (e.g., OO diet) can lead to aberrant lipid metabolism and low antioxidant status in tilapia [24,25]. Therefore, it is imperative to conduct a comprehensive investigation into the metabolic response of tilapia to OO in their diet. Previous investigations have demonstrated that VE supplied at a dose of 50–100 mg/kg is sufficient to satisfy the growth demand of Nile tilapia when the dietary lipid level is set at 50 g/kg [26]. Notably, higher doses of dietary VE have been reported to enhance the immune response [18] and stimulate ovarian development [27,28] in tilapia. Based on these findings, the objective of this study was to elucidate the mechanisms underlying the effects of dietary OO on lipid metabolism, antioxidant capacity, and ovarian development in Nile tilapia (*Oreochromis niloticus*). We also aimed to investigate the potential mitigating effects of vitamin E (VE). The findings of this study will serve as a valuable reference for the application and development of OO-based feed for use in tilapia aquaculture.

## 2. Materials and Methods

### 2.1. Experimental Animals

Female tilapia were acquired from the YangZhong Pilot Research Station located in YangZhong, China. Upon acquisition, the fish were acclimated in polyethylene tanks equipped with a recirculating system (temperature 27–29 °C; dissolved oxygen > 5 mg/L; pH 7.2–7.6) for 1 week. The fish were fed with a commercial feed containing 33% protein and 6% lipid twice per day.

### 2.2. Experimental Diet Preparation

Two VE levels, 70 mg/kg and 200 mg/kg, were selected as experimental variables in this study. Four experimental diets were formulated, incorporating two levels of fat sources and VE, namely: 5% FO with 70 mg/kg VE (hereinafter referred to as 70VE/FO), 5% FO with 200 mg/kg VE (200VE/FO), 5% OO with 70 mg/kg VE (70VE/OO), and 5% OO with 200 mg/kg VE (200VE/OO). The formulation of the experimental diets is listed in Table 1. The dry ingredients were pulverized and sifted through a 60-mesh sieve. A thorough mixing process was carried out using the progressive enlargement principle. Two levels of VE were completely dissolved in experimental oil and then combined with the dry ingredients. To achieve a moist dough, the mixture was supplemented with 8% water content. The dough was processed through a laboratory pelletizer to form pellets with a diameter of 2.5 mm. These pellets were air dried at room temperature and stored at −20 °C until use.

### 2.3. Feeding Trial and Sample Collection

Female Nile tilapia with an initial total weight averaging 162.7 ± 3.8 g were used in this study. After acclimation, the fish were randomly allocated into four dietary groups, which were kept in polyethylene tanks with dimensions of 2080 mm (diameter) × 1200 mm (height). Each tank contained 20 fish. Fish were fed to apparent satiation twice daily (7.30 and 16.30) with the same rearing conditions as those used during the acclimation phase.

The duration of the feeding trial was 10 weeks. Prior to sampling, the fish underwent a 24 h fasting period. Subsequently, the number of experimental fish per tank was recorded, and then all the fish were weighed. From each tank, three fish were randomly chosen and subjected to anesthesia with 100 mg/L MS-222. Blood samples were initially obtained from the caudal vein and promptly centrifuged (3000× *g*, 15 min, 4 °C). The serum samples were collected and stored at −20 °C until serum biochemical analysis. Simultaneously, ovarian tissues were carefully collected and divided into two portions. The first portion was stored at −80 °C for analyses of enzyme activities and mRNA levels. The second portion was stored at −20 °C until determination of fatty acid composition and VE content. Finally, one fish from each tank was chosen to obtain ovarian tissues for histological analyses.

### 2.4. Proximate Composition and VE Content Analysis

The experimental diets and whole fish bodies were analyzed to detect their moisture, crude protein, fat, and ash contents. The analyses followed the procedures outlined by the AOAC [29].

The vitamin E contents in the diet and fish tissues were analyzed as described in our previous study [18]. Briefly, approximately 0.1 g sample was homogenized in physiological saline, with a weight-to-volume ratio of 1:9. The mixture was centrifuged at 2500× *g* for 10 min and the supernatant was used for VE content determination by high-performance liquid chromatography (JY/T 024–1996).

### 2.5. Histological Analysis

The collected ovarian tissues were fixed in 4% paraformaldehyde for 24 h, dehydrated using an ethanol gradient, and then embedded in paraffin. The samples were sectioned into 4 μm slices, followed by staining with hematoxylin and eosin. Pathological changes within the ovarian tissues were visualized and examined under a light microscope (Leica UB203I, Nussloch, Germany).

### 2.6. Serum Hormone Contents and Inflammatory Parameters

The levels of follicle stimulating hormone (FSH), luteinizing hormone (LH), and estradiol (E2) in the serum samples were determined using ELISA methods [30]. The ELISA kits for these analyses were obtained from Shanghai Langdon Biotechnology Co., Ltd. (Shanghai, China), and the manufacturer’s instructions were followed. The activities of serum alanine aminotransferase (ALT) and aspartate transaminase (AST) were measured using an automatic biochemical analyzer (bs-400, MINDRAY, Shenzhen, China). The reagents for these measurements were obtained from Shenzhen Mindray Bio-Medical Electronics Co., Ltd. (Shenzhen, China).

### 2.7. Antioxidant Parameters 

The activities of superoxide dismutase (SOD) and catalase (CAT) as well as the malondialdehyde (MDA) content in both serum and ovarian samples were assessed using established methods [31,32]. The kits used for these measurements were obtained from Nanjing Jiancheng Bioengineering Institute (Nanjing, China).

### 2.8. Lipid Extraction and Analysis

Total lipids were extracted from ovarian tissues following Folch’s method [33]. The levels of ovarian triglycerides (TG) and total cholesterol (TC) were determined using kits obtained from the Nanjing Jiancheng Bioengineering Institute.

Fatty acid composition was analyzed as described elsewhere [34]. Briefly, all lipids were initially separated using a chloroform/methanol mixture (2:1). Subsequently, the lipids were methylated in a 1:99 sulfuric acid:methanol solution at 70 °C for 3 h to generate fatty acid methyl esters (FAMEs). The FAMEs were then extracted using heptane and subjected to gas chromatography analysis utilizing a GC-2010 instrument (Shimadzu, Kyoto, Japan). Fatty acids were identified by comparison with known standards obtained from Sigma (St Louis, MO, USA) and quantified using the CLASS-GC10 GC workstation (Shimadzu). The fatty acid composition of the experimental diets is presented in Table 2.

### 2.9. RNA Extraction and qRT-PCR Analysis 

Total RNA was extracted from ovarian tissues using TRIZOL reagent (Invitrogen, Carlsbad, CA, USA). The cDNA was synthesized with PrimeScript RT Master Mix (Takara, Dalian, China). Quantitative real-time PCR (qPCR) was conducted on an ABI QuantStudio 5 instrument (ABI, Foster City, CA, USA) utilizing the SYBR^®^ Premix Ex Taq kit (Takara). All reactions were performed in triplicate. Elongation factor 1α (*ef1α*) and *β-actin* were utilized as reference genes. The relative transcript levels of the target genes were calculated using the 2^^(-∆∆Ct)^ method, where ∆Ct = Ct target − (Ct EF1α + Ct β-actin)/2. The sequences of primers used for qPCR analyses are listed in Appendix A.

### 2.10. Calculations and Statistical Analysis

The weight gain rate (WGR), specific growth rate (SGR), survival rate (SR), hepatosomatic index (HSI), and GSI were calculated using standard formulae [16].

All results shown in figures and tables are mean ± standard error. The data were tested for normality and homogeneity of variance using the Shapiro–Wilk’s test and Levene’s test, respectively. Factorial (two-way) analysis of variance (ANOVA) was employed to assess the effects of dietary lipid source, VE levels, and their interactions on growth performance, lipid metabolism, and the antioxidant response. Significance was determined at *p* < 0.05. If significant differences were found, further analyses were conducted using one-way ANOVA to assess the effects of the dietary lipid source and VE level for each respective factor. 

## 3. Results

### 3.1. Growth Performance and Whole-Body Composition

The growth performance of fish fed with the different diets is shown in Figure 1. The FBW, WGR, SGR, HSI, and GSI were significantly influenced by the dietary lipid level (*p* < 0.05), and GSI was significantly affected by the interaction between dietary lipid and VE levels (*p* < 0.05). Specifically, FBW, WGR, SGR, and GSI were significantly lower in the 70VE/OO treatment than in the 70VE/FO treatment (*p* < 0.05), while HSI was significantly higher in the 70VE/OO treatment than in the 70VE/FO treatment (*p* < 0.05). For fish fed with OO-containing diets, GSI significantly increased with increasing dietary VE levels (*p* < 0.05). 

The proximate composition of whole fish and the VE content in tilapia fed with the different experimental diets are shown in Table 3. Ovarian VE contents were significantly higher in the 200VE groups than in the 70VE groups (*p* < 0.05). However, moisture, crude protein, crude lipid, and ash contents of whole fish bodies were not significantly affected by the dietary lipid or VE levels (*p* > 0.05).

### 3.2. Ovary Morphology

Figure 2 shows various stages of oocyte development in female tilapia fed with different diets. In the 70VE/OO group, the majority of oocytes were at stage II and III, and atretic follicles were present. Conversely, in the 70VE/FO group, some oocytes had progressed to stage V, although a certain number of oocytes were still at stages II and III. The oocytes in the 200VE group reached a more advanced stage of maturity than did those in the 70VE group. There were more stage III and IV oocytes in the 200VE/OO group than in the 70VE/OO group and more stage V oocytes in the 200VE/FO group than in the 70VE/FO group.

### 3.3. Serum Hormone Contents and Inflammatory Parameters

The serum hormone contents and inflammatory parameters of female tilapia fed with different diets are shown in Figure 3. The serum E2, FSH, LH, and AST contents were significantly affected by the dietary lipid level (*p* < 0.05), and the serum E2, LH, AST, and ALT contents were significantly affected by the dietary VE level (*p* < 0.05). The serum AST content was also significantly influenced by the interaction between dietary lipid and VE levels (*p* < 0.05). In the 70VE groups, compared with dietary FO, dietary OO resulted in significantly decreased serum E2, FSH, and LH levels and significantly increased serum AST and ALT levels (*p* < 0.05). In the fish fed with OO diets, serum E2 and LH levels significantly increased, while AST and ALT levels significantly decreased with increasing dietary VE levels (*p* < 0.05). In the 200VE groups, compared with dietary FO, dietary OO significantly increased serum AST levels (*p* < 0.05).

### 3.4. Serum and Ovary Antioxidant Parameters 

The effects of the different diets on serum and ovarian antioxidant parameters are shown in Figure 4. The serum and ovarian antioxidant parameters were significantly influenced by the dietary lipid level (*p* < 0.05), and the serum and ovarian MDA contents were significantly affected by the dietary VE level (*p* < 0.05). In the 70VE groups, compared with dietary FO, dietary OO resulted in a significant decrease in the activities of SOD and CAT, and significant increases in the contents of MDA in both serum and ovarian tissues (*p* < 0.05). Furthermore, in the fish fed with OO diets, the serum SOD activity and ovarian SOD and CAT activities significantly increased and the serum and ovarian MDA contents significantly decreased with increasing dietary VE levels (*p* < 0.05). Conversely, in the 200VE groups, compared with dietary FO, dietary OO resulted in significantly lower serum SOD activity (*p* < 0.05).

### 3.5. Lipid Content and Fatty Acid Composition in the Ovary

The TG and TC contents in the ovary were significantly affected by dietary lipid levels (*p* < 0.05), and the TG content in the ovary was also significantly affected by the dietary VE levels (*p* < 0.05) (Figure 5A,B). In the 70VE groups, compared with dietary FO, dietary OO significantly decreased ovarian TG and TC contents (*p* < 0.05). In the fish fed with OO-containing diets, the ovarian TG content significantly increased with increasing dietary VE levels (*p* < 0.05).

The ovarian fatty acid composition in fish fed with the four experimental groups is shown in Appendix A and Figure 5C. The contents of SFA, MUFA, PUFA, n-3 PUFA, C20:4 (ARA), C20:5 (EPA), and C22:6 (DHA) in the ovary were significantly affected by the dietary lipid level (*p* < 0.05), and the contents of MUFA, PUFA, n-3 PUFA, n-6 PUFA, C20:4, C20:5, and C22:6 in the ovary were significantly affected by the dietary VE level (*p* < 0.05). The C20:4 content in the ovary was significantly affected by the interaction between dietary lipid and VE levels (*p* < 0.05) (Figure 5D–K). Compared with the 70VE groups, the 200VE groups showed significantly increased ovarian SFA, PUFA, n-3 PUFA C20:4, C20:5, and C22:6 contents, and decreased hepatic MUFA content (*p* < 0.05). In the fish fed with OO-containing diets, the hepatic PUFA, n-6 PUFA, C20:4, and C22:6 contents significantly increased with increasing dietary VE levels (*p* < 0.05).

### 3.6. Expression Levels of Genes Encoding Hormone Receptors in the Ovary

The transcript levels of *erα*, *fshr*, and *lhr* in the ovary were significantly affected by both dietary lipid levels and VE levels (*p* < 0.05), and the transcript level of *lhr* in the ovary was also significantly affected by the interaction between dietary lipid and VE levels (*p* < 0.05) (Figure 6). In the 70VE groups, compared with dietary FO, dietary OO significantly decreased the transcript levels of *erα*, *fshr*, and *lhr* in the ovary (*p* < 0.05). In the fish fed with OO diets, the transcript levels of ovarian *erα*, *erβ*, *fshr* and *lhr* significantly increased with increasing dietary VE levels (*p* < 0.05). In the 200VE groups, compared with dietary FO, dietary OO significantly decreased the transcript levels of *fshr* and *lhr* in the ovary (*p* < 0.05).

### 3.7. Expression Levels of Genes Related to Lipid Metabolism in the Ovary

The transcript levels of *pparγ*, *hsl*, *accα*, and *elovl6* in the ovary were significantly affected by the dietary lipid level (*p* < 0.05), and that of *pparα* was significantly affected by the dietary VE level (*p* < 0.05). The transcript levels of *pparα*, *accα*, and *elovl6* were significantly affected by the interaction between dietary lipid and VE levels (*p* < 0.05) (Figure 7). In the 70VE groups, compared with dietary FO, dietary OO significantly decreased the transcript levels of *pparα*, *accα*, *elovl6*, and *hsl*, and significantly increased the transcript level of *pparγ* in the ovary (*p* < 0.05). In the fish fed with OO-containing diets, the transcript levels of *pparα*, *accα* and *elovl6* in the ovary significantly increased with increasing dietary VE levels (*p* < 0.05). In the 200VE groups, compared with dietary FO, dietary OO significantly decreased the transcript levels of *accα* and *elovl6* in the ovary (*p* < 0.05).

### 3.8. Transcript Levels of Genes Related to the nrf2 Signaling Pathway in the Ovary

The transcript levels of *nrf2* and *keap1* in the ovary were significantly affected by the dietary lipid level (*p* < 0.05), and that of *keap1* was significantly affected by the dietary VE level (*p* < 0.05) (Figure 8). In the 70VE groups, compared with dietary FO, dietary OO significantly decreased the transcript level of *nrf2* and decreased that of *keap1* in the ovary (*p* < 0.05). In the fish fed with OO-containing diets, the transcript levels *nrf2* increased and those of *keap1* decreased with increasing dietary VE levels (*p* < 0.05). 

### 3.9. Expression Levels of Genes Related to nf-κb Signaling Pathway in the Ovary

The transcript levels of four *nf-κb* signaling pathway-related genes were significantly affected by the dietary lipid level (*p* < 0.05), the transcript level of *il1β* was significantly affected by the dietary VE levels (*p* < 0.05), and the transcript levels of *nf-κb tnfα*, and *il1β* were also significantly affected by the interaction between dietary lipid and VE levels (*p* < 0.05) (Figure 9). In the 70VE groups, compared with dietary FO, dietary OO significantly increased the transcript levels of *nf-κb*, *infγ*, *tnfα*, and *il1β* in the ovary (*p* < 0.05). In the fish fed with OO-containing diets, the transcript levels of *nf-κb* and *il1β* in the ovary significantly decreased with increasing dietary VE levels (*p* < 0.05). In the 200VE groups, compared with dietary FO, dietary OO significantly increased the transcript level *il1β* in the ovary (*p* < 0.05).

## 4. Discussion

Traditionally, FO has been the main lipid source used in aquafeeds. However, the aquaculture industry is facing a shortage of FO due to its increasing price and limited production. Vegetable oil is considered an excellent substitute for FO because of its cost-effectiveness and wide availability [2]. OA-rich OO has been used successfully as a substitute for FO in feed for some aquatic animals [4,35]. Notably, excessive replacement of dietary FO with OO can result in lipid metabolism disorders and decreased antioxidant capacity, which can adversely affect fish growth and overall health. For example, Li et al. [3] reported that complete replacement of FO with OO in the diet led to decreases in the values of growth indicators such as FBW and SGR in large yellow croaker. Similarly, juvenile Nile tilapia fed with an OO-containing diet showed poorer growth performance than that of those fed with a FO-containing diet [25]. In the present study, the values of FBW, WGR, and SGR were lower in fish fed with an OO-containing diet than in those fed with a FO-containing diet, indicating that excessive replacement of dietary FO with OO adversely affected the growth of female tilapia. Furthermore, the HSI was higher in the OO groups than in the FO groups, similar to large yellow croaker fed with an OO-containing diet [3]. Our results suggest that excessive replacement of dietary FO by OO disrupts lipid metabolism, resulting in hepatic lipid accumulation and elevated HSI. 

In the present study, the dietary VE level did not significantly affect the growth performance of female tilapia. In studies on spotted seabass (*Lateolabrax macroatus*) [36], golden pompano [37], and blunt snout bream (*Megalobrama ambycephala*) [38], appropriate VE supplementation was found to have a growth-promoting effect, although high doses of VE did not further increase fish growth. Previous studies have shown that 50–100 mg/kg VE can meet the growth demands for Nile tilapia at a dietary lipid level of 50 g/kg [18]. Our results show that 70 mg/kg VE in the diet was able to meet the growth demands of female tilapia, while VE supplementation at a high dose (200 mg/kg) did not further enhance growth performance.

Gonad development in fish is mainly influenced by environmental conditions and feed nutrition. Research has shown that high-quality protein and lipid sources, as well as appropriate vitamin levels, can improve reproductive performance and accelerate oocyte development in fish [37,39]. Studies on tongue sole [8], Atlantic Salmon [9], and sterlet sturgeon [10] have shown that partially replacing FO with OA-enriched OO or RO can improve the egg quality and reproductive performance of broodstock. In this study, the majority of the oocytes in fish fed with an OO-containing diet were at developmental stages II and III, consistent with the lower GSI. The GSI serves as a dependable indicator of the gonads’ developmental status [40]. Our results show that an OO-containing diet inhibited ovarian development in female tilapia to a certain extent, but VE supplementation at 200 mg/kg in the OO-containing diet increased the GSI and the number of stage III and IV oocytes. In other studies, appropriate VE supplementation was shown to promote the reproductive performance of red swordtail (*Xiphophorus helleri*) [16] and Black Sea trout (*Salmo labrax*) [17]. The results of our study suggest that VE at 70 mg/kg in the OO-containing diet was sufficient to meet the growth requirements of female tilapia, but insufficient to meet the demands for ovarian development. Extra VE supplementation (at 200 mg/kg) promoted the maturation of oocytes and improved the gonad development of female tilapia fed with an OO-containing diet. A study conducted on Nile tilapia revealed that the most favorable reproductive performance was observed in fish fed a diet containing soybean oil supplemented with 120 mg/kg of VE [27]. Another study demonstrated that a VE supplementation of 400 mg/kg effectively met the reproduction requirements for female Nile tilapia when the dietary lipid level was set at 100 g/kg [28]. The variation in VE requirements for gonad development and reproduction among above studies may be attributed to factors such as tilapia size, age, lipid levels, and types of lipids used.

Sex hormones play crucial roles in gonad differentiation and development in teleosts [41]. The main estrogen subtype, E2, is involved in the regulation of reproductive physiology, ovarian differentiation, and oocyte development [42]. One study detected a positive correlation between the E2 level and yolk accumulation in oocytes [43]. In the current study, the elevated serum E2 level in female tilapia fed with the FO-containing diet appeared to be associated with accelerated yolk accumulation in oocytes. However, feeding with the OO-containing diet resulted in a significant decrease in serum E2 levels, which may have hindered oocyte development. The pituitary gland secretes the gonadotropins LH and FSH, which regulate gonad development and sex hormone synthesis [44]. Insufficient FSH secretion has been reported as the primary cause of follicular atresia [23]. In addition, Qiang et al. reported that the serum LH and FSH levels are closely related to gonad development in tilapia, and their levels gradually increase during ovarian development and peak when oocytes mature [30]. In this study, compared with the FO groups, the groups fed with the OA-enriched OO diet showed decreased levels of serum E2, LH, and FSH, as well as lower transcript levels of *erα*, *fshr*, and *lhr*, which encode hormone receptors, in the ovary. Supplementation with OA was shown to decrease the transcript levels of *fshr* and *lhr* and diminish steroid hormone production in bovine granulosa cells [45]. We speculate that a diet enriched with OA from OO may impede the synthesis and secretion of E2, LH, and FSH. Furthermore, it may decrease the sensitivity of the ovary to gonadotropins, resulting in the inhibition of ovarian development and accelerated onset of follicular atresia. Nevertheless, we found that VE supplementation at 200 mg/kg in the OO-containing diet increased the serum sex hormone levels and the transcript levels of genes encoding their receptors in the ovary. Similar results have been reported in aging breeder hens [46], indicating that VE supplementation may help to regulate follicular development by enhancing sex hormone biosynthesis.

Lipids are essential nutrients for fish gonad development, mating reproduction, parental fertility, and embryonic development [47]. As the ovaries develop, they gradually accumulate lipids, with lipid content peaking at ovary maturity [48,49]. In the current study, the TG and total cholesterol TC levels in the ovary were higher in the 70VE/FO group than in the 70VE/OO group. Similar results have been reported for spotted scat (*Scatophagus argus*) [50] and gilthead seabream [7]. Based on the changes in lipid indicators in this experiment, we speculate that the inclusion of OO in the diet may impede the transport of lipids from the liver to the ovaries. This, in turn, could result in elevated HSI and reduced lipid accumulation in the ovaries, conditions that are not conducive to successful oocyte maturation. In this experiment, dietary OO significantly increased the proportion of MUFA (especially C18:1), but significantly decreased the proportion of n-3 PUFA in the ovary. Similar findings were reported in the analysis of the body fatty acid composition of Nile tilapia [25]. This suggests that the composition of fatty acids in tilapia tissues maybe greatly affected by the type and quantity of lipids in the diet. Several studies have noted that the accumulation of n-3 PUFA, especially C20:5n-3 (EPA) and C22:6n-3 (DHA), improves the egg quality and reproductive performance of broodstock [51,52], while the overaccumulation of C18:1 is not conducive to sex hormone synthesis or the normal development of germ cells [45]. Our results suggest that dietary OO caused an imbalance in fatty acid composition and some essential fatty acids did not accumulate to sufficient levels in the ovary. This is one of the possible reasons why ovarian development was inhibited in female tilapia fed with an OO-containing diet. Our results also show that VE supplementation at 200 mg/kg in the OO-containing diet increased the TG content and the proportion of LC-PUFA out of total fatty acids in the ovary. Dietary VE supplementation was also shown to promote TG accumulation in the ovary of aging breeder hens [46]. In that study, it was considered that the accumulated TG may promote the production of yolk precursors in the ovary, thereby promoting the oocyte development. At the same time, VE can effectively reduce the peroxidation of unsaturated fatty acids because of its antioxidant properties [12]. Studies on blunt snout bream [38] and golden pompano [37] reported that n-3 PUFA, especially EPA and DHA, accumulated to higher levels in tissues with increasing dietary VE supplementation. Thus, the extra VE supplementation in the OO-containing diet may have contributed to lipid accumulation, regulation of the composition of fatty acids, and improvements in the lipid nutritional status for ovarian development in female tilapia.

As the main lipid in oocytes, TG can be hydrolyzed by lipases to generate fatty acids, which are further decomposed by β-oxidation to provide energy [53]. Thus, abnormal lipid metabolism can affect the development and maturation of oocytes. Generally, the regulation of lipid metabolism relies on multiple lipid metabolism-related enzymes and transcription factors [54]. The peroxisome proliferator-activated receptor family are ligand-dependent transcription factors. One of its members, encoded by *pparα*, plays an important role in regulating fatty acid β-oxidative metabolism, while another member, encoded by *pparγ*, is mainly involved in lipid biosynthesis [55]. HSL is a rate-limiting enzyme for TG decomposition [56]. The transcript levels of *pparα* and *lpl* were found to be upregulated during follicular development of spotted scat [57]. Wang et al. reported that the expression level of ovarian *pparγ* may be regulated by dietary n-3 PUFA levels, and a low expression level of *pparγ* was beneficial for oocyte maturation in spotted scat [50]. In this study, compared with the 70VE/FO group, the 70VE/OO group showed significantly lower transcript levels of *pparα* and *hsl* and significantly higher transcript levels of *pparγ* in the ovary, indicating that dietary OO inhibited TG, hydrolysis, and fatty acid β-oxidation, possibly resulting in insufficient energy for oocyte development in female tilapia. ACCα and ELOVL6 are key enzymes for *de novo* synthesis of fatty acids [58,59]. In this study, the transcript levels of ovarian *accα* and *elovl6* were downregulated in fish fed with an OO-containing diet. This suggests that feed containing OO not only inhibits fatty acid β-oxidation but also hinders lipid synthesis, consistent with our histological observations. The dysregulation of lipid metabolism in the ovary may have contributed to the poor ovarian development. Previous studies have shown that VE can regulate lipid metabolism by modulating the expression of lipid metabolism-related genes (e.g., *ppars, elovls, fas*) in fish [37,38]. In this study, VE supplementation at 200 mg/kg in the OO-containing diet increased the transcript levels of *pparα*, *hsl*, *accα* and *elovl6*, indicating that extra VE supplementation may regulate fatty acid metabolism via activating the *pparα* signaling pathway, thereby enabling oocytes to obtain more energy for development.

Metabolic processes result in the production of ROS in the fish body. Environmental stress and nutrient imbalances in the diet can result in excess ROS, which cause oxidative damage to cells [60]. The *nrf2* signaling pathway plays a crucial role in mitigating oxidative stress damage and has been extensively investigated in both mammals and fish [61]. Under oxidative challenge, *nrf2* activates a cascade of downstream genes and enzymes related to antioxidant defense, such as SOD and CAT, which work together to reduce the detrimental effects of oxidative stress [62]. Severe oxidative stress can inhibit the *nrf2* signaling pathway [63]. In this study, the transcript level of *nrf2* and the activities of SOD and CAT were significantly downregulated, while the content of MDA was significantly upregulated, in the 70VE/OO group compared with the 70VE/FO group. These results suggest that dietary OO reduces the antioxidant capacity in female tilapia. Similar results were observed in tilapia [24], large yellow croaker [3], and Japanese sea bass (*Lateolabrax japonicus*) [64]. Aquatic organisms require a large amount of energy to activate and maintain antioxidant systems to cope with oxidative stress [65]. We speculate that dietary OO may inhibit fatty acid β-oxidation, thereby limiting the energy supply in the ovary, leading to the inactivation of the *nrf2* signaling pathway and weakening of the antioxidant response. Ultimately, these changes aggravate oxidative damage to oocytes in female tilapia. Appropriate VE supplementation has been shown to improve the antioxidant capacity and immunity of fish [13,18]. For example, Qiang et al. found that moderate supplementation of dietary VE activated antioxidant genes in genetically improved farmed tilapia, leading to the enhancement of antioxidant capacity [18]. In this study, VE at 200 mg/kg in the OO diet upregulated *nrf2*, increased the activities of SOD and CAT, and reduced the MDA content in the ovary. This may be related to VE blocking the production of peroxides by participating in redox reactions with oxygen-containing groups of ROS [66]. Dietary supplementation with VE has been shown to have an ameliorating effect on oxidative stress in aquatic organisms caused by metal pollution [19], oxidized fish oil [20], and high-fat feed [21]. Our results suggest that VE supplementation in an OO-containing diet may improve the oxidative stress status of female tilapia by activating the *nrf2* signaling pathway.

The *nf-κb* signaling pathway participates in the regulation of the inflammatory response in fish. Stimulation by exogenous or endogenous stresses activates the *nf-κb* signaling pathway in fish, and the resulting release of inflammatory cytokines leads to inflammation [67]. In this study, compared with the 70VE/FO group, the 70VE/OO group showed higher transcript levels of *nf-κb* and its downstream genes *infγ*, *tnfα*, *il1β* encoding inflammatory factors. Combined with the changes in serum AST and ALT levels, these findings suggest that dietary OO causes inflammation as well as oxidative stress in female tilapia. Studies on large yellow croaker showed that, compared with an FO group, an OO group showed increased expression of pro-inflammatory genes, *cox-2*, *il-1β*, and *tnfα*, which ultimately induced inflammation [3]. Similar findings have been reported in rats [68]. The occurrence of inflammation is closely related to ovarian dysplasia and follicular atresia [69]. In this experiment, dietary OO aggravated inflammation in the ovary, and this may be one of the reasons why ovarian dysplasia occurred. Dietary VE was shown to alleviate inflammation in olive flounder (*Paralichthys olivaceus*) [70]. In the present study, VE supplementation in the OO-containing diet reduced the transcript levels of *nf-κb* and *il1β*, suggesting that inflammation in the ovary was alleviated to some extent, and this may have been conducive to oocyte maturation in female tilapia.

## 5. Conclusions

Dietary OO negatively affected ovarian development and growth performance in female tilapia by reducing the contents of sex hormones, disrupting lipid metabolism, and increasing oxidative stress and inflammation (Figure 10). However, VE supplementation at a dose of 200 mg/kg in the OO-containing diet improved the antioxidant capacity of female tilapia and alleviated inflammation and the disruption of lipid metabolism (Figure 10). These effects may have been due to the antioxidant properties of VE and its ability to regulate the *nrf2*-*nf-κb* signaling pathway. Overall, the results of this study provide a reference for improving the utilization efficiency of OO in feed for farmed tilapia.

## Figures and Tables

**Figure 1 antioxidants-12-01524-f001:**
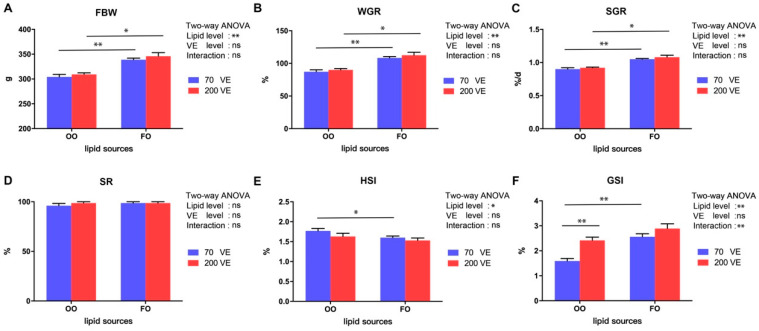
Growth performance and gonadal development of female tilapia fed with different dietary lipid and VE levels for 10 weeks. (**A**) FBW; (**B**) WGR; (**C**) SGR; (**D**) SR; (**E**) HSI; (**F**) GSI. **: *p* < 0.01, *: *p* < 0.05.

**Figure 2 antioxidants-12-01524-f002:**
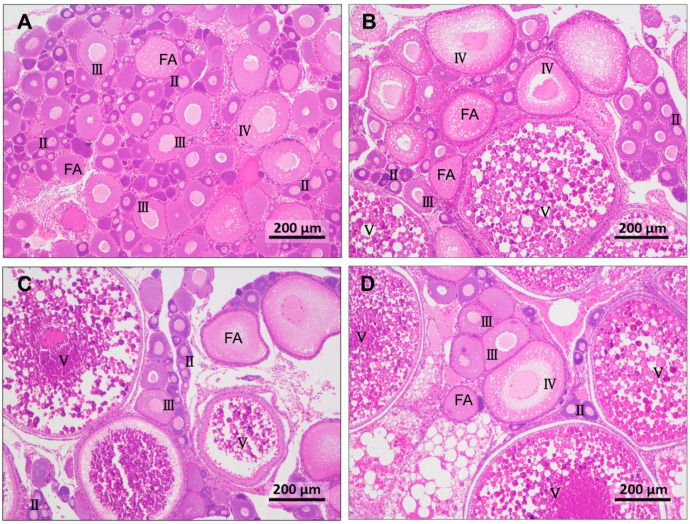
Hematoxylin and eosin-stained gonad tissue from female tilapia fed with different dietary lipid and VE levels for 10 weeks (magnification ×100, scale bar: 200 μm). (**A**) oocyte statue of fish fed with 70VE/OO diet; (**B**) oocyte statue of fish fed with 200VE/OO diet; (**C**) oocyte statue of fish fed with 70VE/FO diet; (**D**) oocyte statue of fish fed with 200VE/FO diet; II, III, IV, and V represent oocytes at stages II, III, IV, and V, respectively; FA: follicular atresia.

**Figure 3 antioxidants-12-01524-f003:**
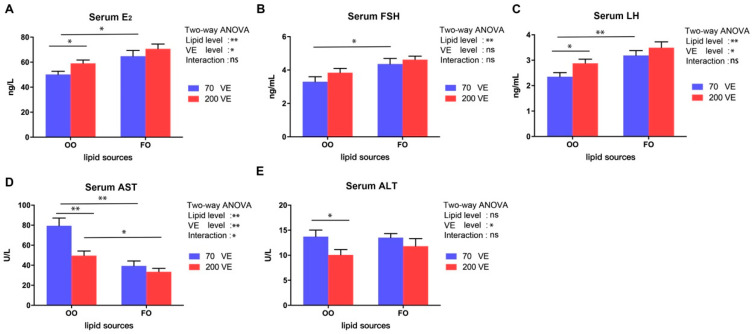
Serum hormone and inflammatory parameters of female tilapia fed with different dietary lipid and VE levels for 10 weeks (*n* = 9). (**A**) E2; (**B**) FSH; (**C**) LH; (**D**) AST; (**E**) ALT. **: *p* < 0.01, *: *p* < 0.05.

**Figure 4 antioxidants-12-01524-f004:**
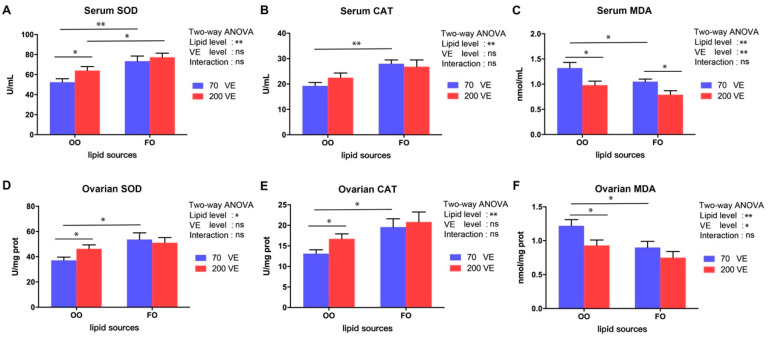
Serum and ovarian oxidative status of female tilapia fed with different dietary lipid and VE levels for 10 weeks (*n* = 9). (**A**) serum SOD; (**B**) serum CAT; (**C**) serum MDA; (**D**) ovarian SOD; (**E**) ovarian CAT; (**F**) ovarian MDA. **: *p* < 0.01, *: *p* < 0.05.

**Figure 5 antioxidants-12-01524-f005:**
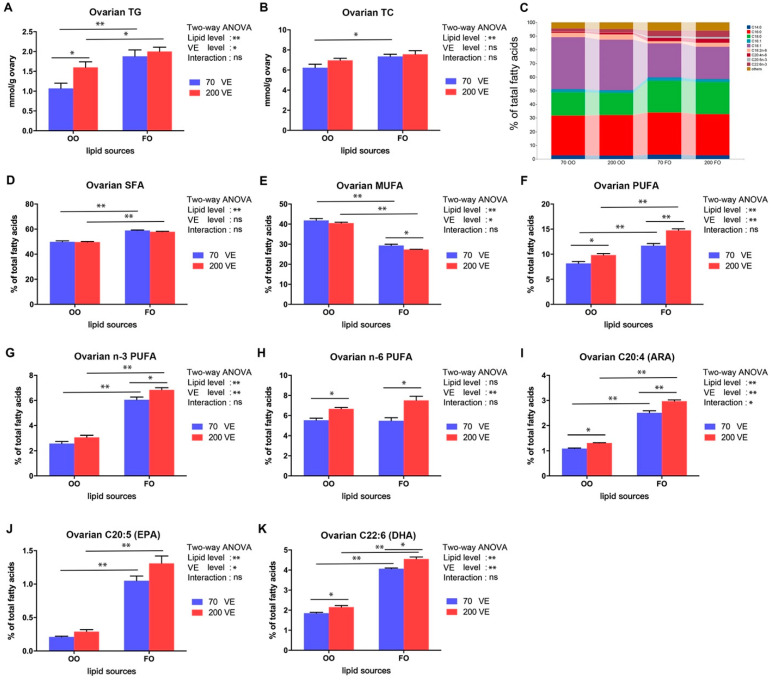
Ovarian triglyceride, and total cholesterol and fatty acid composition of female tilapia fed with different dietary lipid and VE levels for 10 weeks (*n* = 9). (**A**) TG; (**B**) TC; (**C**) expression profile of the top 10 fatty acids in the ovary; (**D**) SFA; (**E**) MUFA; (**F**) PUFA; (**G**) n-3 PUFA; (**H**) n-6 PUFA; (**I**) ARA; (**J**) EPA; (**K**) DHA. **: *p* < 0.01, *: *p* < 0.05.

**Figure 6 antioxidants-12-01524-f006:**
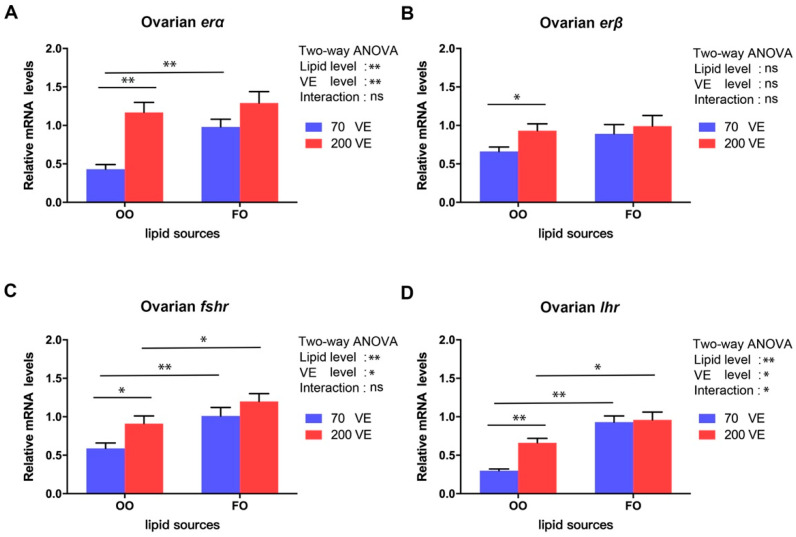
Expression levels of genes related to hormone receptors in ovary of female tilapia fed with different dietary lipid and VE levels for 10 weeks (*n* = 9). (**A**) *erα*; (**B**) *erβ*; (**C**) *fshr*; (**D**) *lhr*. **: *p* < 0.01, *: *p* < 0.05.

**Figure 7 antioxidants-12-01524-f007:**
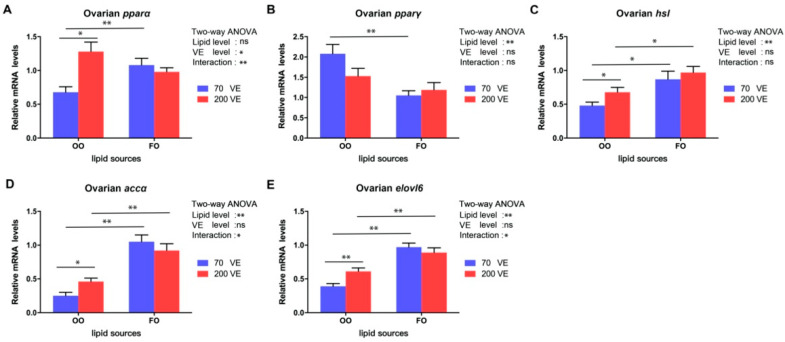
Expression levels of genes related to lipid metabolism in ovary of female tilapia fed with different dietary lipid and VE levels for 10 weeks (*n* = 9). (**A**) *pparα*; (**B**) *pparγ*; (**C**) *hsl*; (**D**) *accα*; (**E**) *elovl6.* **: *p* < 0.01, *: *p* < 0.05.

**Figure 8 antioxidants-12-01524-f008:**
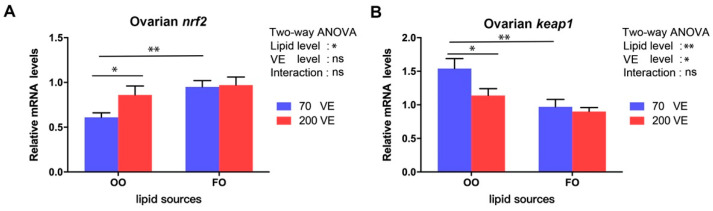
Expression levels of genes related to *nrf2* signaling pathway in ovary of female tilapia fed with different dietary lipid and VE levels for 10 weeks (*n* = 9). (**A**) *nrf2*; (**B**) *keap1*. **: *p* < 0.01, *: *p* < 0.05.

**Figure 9 antioxidants-12-01524-f009:**
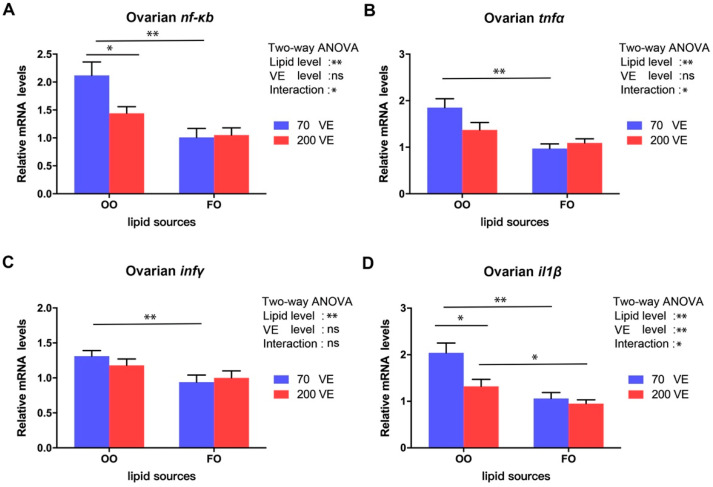
Expression levels of genes related to *nf-κb* signaling pathway in ovary of female tilapia fed with different dietary lipid and VE levels for 10 weeks (*n* = 9). (**A**) *nf-κb*; (**B**) *tnfα*; (**C**) *infγ*; (**D**) *il1β*. **: *p* < 0.01, *: *p* < 0.05.

**Figure 10 antioxidants-12-01524-f010:**
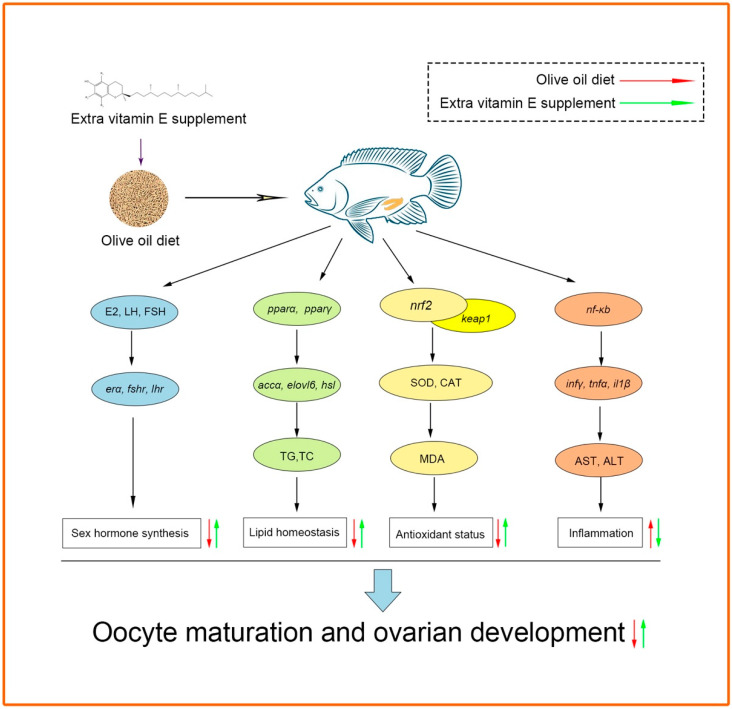
Possible regulation mechanisms of ovarian development in female tilapia in response to dietary high fat and VE.

**Table 1 antioxidants-12-01524-t001:** Experimental feed formula and nutrient composition (g/1000 g).

	70 OO	200 OO	70 FO	200 FO
Fish meal	60.00	60.00	60.00	60.00
Wheat middling	80.00	80.00	80.00	80.00
Wheat flour	200.00	200.00	200.00	200.00
Olive oil	50.00	50.00		
Fish oil			50.00	50.00
Soybean meal	200.00	200.00	200.00	200.00
Cottonseed meal	180.00	180.00	180.00	180.00
Rapeseed meal	165.00	165.00	165.00	165.00
Ca(H_2_PO_4_)_2_	15.00	15.00	15.00	15.00
α-cellulose	32.93	32.8	32.93	32.8
Vitamin Premix (vitamin E free) ^‡^	5.00	5.00	5.00	5.00
Mineral Premix ^§^	5.00	5.00	5.00	5.00
Vitamin C phosphate ester	2.00	2.00	2.00	2.00
Choline chloride	5.00	5.00	5.00	5.00
Vitamin E	0.07	0.20	0.07	0.20
Total	1000	1000	1000	1000
Proximate composition (% dry matter)				
Dry matter	94.79	94.81	94.72	94.93
Crude protein	33.82	33.92	33.94	33.89
Crude lipid	6.64	6.83	6.75	6.73
Ash	4.55	4.51	4.57	4.49
V_E_ (mg/kg)	70.75	199.87	70.32	200.22

‡ Vitamin premix (mg/kg dry diet):V_A_ 10, V_D_ 0.05, V_K_ 40, V_B1_ 50, V_B2_ 200, V_B3_ 500, V_B6_ 50, V_B7_ 5, V_B11_ 15, V_B12_ 0.1, V_C_ 1000, inositol 2000, choline 5000. § Mineral premix (mg/kg dry diet): FeSO_4_·7H_2_O 372, CuSO_4_·5H_2_O 25, ZnSO_4_·7H_2_O 120, MnSO_4_·H_2_O 5, MgSO_4_ 2475, NaCl 1875, KH_2_PO_4_ 1000, Ca (H_2_PO_4_)_2_ 2500.

**Table 2 antioxidants-12-01524-t002:** Fatty acid composition of the experimental diets (% of total fatty acids).

Fatty Acid	70 OO	200 OO	70 FO	200 FO
C12:0	0.61	0.57	0.12	0.11
C14:0	0.65	0.57	2.99	2.87
C15:0	0.06	0.06	0.36	0.35
C16:0	14.17	14.10	16.74	16.71
C17:0	0.12	0.15	0.27	0.27
C18:0	3.76	3.70	3.66	3.66
C20:0	0.42	0.42	0.71	0.71
C22:0	0.21	0.21	0.44	0.45
C16:1	0.81	0.84	2.70	2.69
C18:1	56.07	56.11	18.16	18.53
C20:1	0.33	0.35	3.00	3.02
C22:1	0.19	0.28	10.77	11.03
C18:2n-6	16.82	16.68	20.36	20.40
C18:3n-3	1.38	1.38	3.49	3.52
C18:3n-6	0.70	0.71	1.68	1.47
C20:2n-6	0.08	0.09	0.22	0.22
C20:3n-6	0.03	0.03	0.05	0.05
C20:3n-3	0.05	0.06	0.12	0.12
C20:4n-6 (ARA)	0.08	0.10	2.63	2.58
C20:5n-3 (EPA)	0.54	0.56	4.09	3.97
C22:3	1.67	1.77	0.19	0.20
C22:4n-6	0.21	0.20	0.29	0.30
C22:5n-3	0.14	0.13	0.65	0.65
C22:6n-3(DHA)	0.90	0.92	6.31	6.13
ΣSFA	20.00	19.78	25.29	25.13
ΣMUFA	57.40	57.58	34.63	35.27
ΣPUFA	22.60	22.63	40.08	39.61
Σn-3 PUFA	3.01	3.05	14.66	14.39
Σn-6 PUFA	17.92	17.81	25.23	25.02

**Table 3 antioxidants-12-01524-t003:** Proximate composition of whole fish and ovarian vitamin E content in female tilapia fed experimental diets for 10 weeks.

	Moisture (%)	Crude Protein (%)	Crude Lipid (%)	Ash (%)	Ovarian Vitamin E (μg/g Wet Tissue)
70 OO	69.17 ± 0.39	15.57 ± 0.41	8.97 ± 0.39	3.19 ± 0.10	25.47 ± 1.64a
200 OO	68.92 ± 0.38	15.89 ± 0.36	8.85 ± 0.31	3.16 ± 0.06	41.90 ± 1.53 b
70 FO	69.74 ± 0.51	16.53 ± 0.44	8.60 ± 0.32	3.31 ± 0.12	23.78 ± 1.15 a
200 FO	69.92 ± 0.46	16.27 ± 0.33	8.21 ± 0.27	3.25 ± 0.07	44.31 ± 2.08 b
Two-way ANOVA:					
lipid level	ns	ns	ns	ns	ns
VE level	ns	ns	ns	ns	**
Interactions	ns	ns	ns	ns	ns

Notes: Data are presented as mean ± SEM (*n* = 6). Means with different lowercases in the same dietary lipid with different VE level are significantly different from each other. **: *p* < 0.01; ns: *p* > 0.05.

## Data Availability

All of the data is included in the article/Appendix A.

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
