# Peer review of "Vitamin E Ameliorates Impaired Ovarian Development, Oxidative Stress, and Disrupted Lipid Metabolism in Oreochromis niloticus Fed with a Diet Containing Olive Oil Instead of Fish Oil"

_antioxidants, 2023, doi:10.3390/antiox12081524_

Round 1
Reviewer 1 Report
This is an interesting manuscript detailing the effects of OO vs FO -based diets on multiple physiological endpoints. However, I am confused about the dietary contents - the paper describes OO vs FO, but both diets contain the same amount of FO. This needs to be explained and further discussed in the context of OO vs FO effects. The paper would be much more interesting and informative if diets contained only OO or only FO. Also, the effects of Vit-E manipulation are interesting outside of the OO vs FO context. It would have been interesting to see the effects of no Vit-E in the diets...or has this already been addressed in other studies? These comments may seem trivial or minor but should be addressed with adequate detail in a revised version of the paper.
Author Response
Dear editors and reviewers,
Thank you for your letter and comments on our manuscript titled "Vitamin E ameliorates impaired ovarian development, oxidative stress, and disrupted lipid metabolism in Oreochromis niloticus fed with a diet containing olive oil instead of fish oil" (antioxidants-2489865). These comments helped us improve our manuscript and provided important guidance for future research.
We have addressed the editor’s and reviewers’ comments to the best of our abilities. We hope our manuscript now meets your requirements for publication.
We marked the revised portions in red in our manuscript. All comments and our specific responses are detailed below.
Reviewer #1: Comments on antioxidants-2489865:
Comment 1: This is an interesting manuscript detailing the effects of OO vs FO -based diets on multiple physiological endpoints. However, I am confused about the dietary contents - the paper describes OO vs FO, but both diets contain the same amount of FO. This needs to be explained and further discussed in the context of OO vs FO effects. The paper would be much more interesting and informative if diets contained only OO or only FO.
Response 1: We have rechecked our experimental feed formula (Table 1) and found that not all feeds have the same amount of fish oil (FO) added. In fact, only the 70VE/FO and 200VE/FO groups added with 5% FO, while the 70VE/OO and 200VE/OO groups added with 5% OO. There may be some errors in the transmission and download process of the manuscript. Please verify Table 1 in our latest revised version. Generally, in aquaculture feed, the majority of dietary lipids are typically obtained from the addition of exogenous oils. Other feed ingredients may also contain small amounts of lipids. In other studies investigating the effects of different lipid sources on the growth and physiological metabolism of tilapia, the feed riched in target oil was also constructed by adding exogenous oil [1, 2]. This suggests that the method of lipid addition utilized in our experiment is acceptable in the tilapia nutritional experiments. Finally, thank you to the reviewer for his/her valuable comments, which will be of great help to our next research work.
Comment 2:Also, the effects of Vit-E manipulation are interesting outside of the OO vs FO context. It would have been interesting to see the effects of no Vit-E in the diets...or has this already been addressed in other studies? These comments may seem trivial or minor but should be addressed with adequate detail in a revised version of the paper.
Response 2: We revised and added more detailed information in our manuscript. The revised part is in page [2], lines [65–68] and lines [71-73].
Reference:
- Peng, X.; Li, F.; Lin, S.; Chen, Y., Effects of total replacement of fish oil on growth performance, lipid metabolism and antioxidant capacity in tilapia (Oreochromis niloticus). Aquacult. Int. 2016, 24, (1), 145-156.
- Apraku, A.; Huang, X.; Ayisi, C. L., Effects of alternative dietary oils on immune response, expression of immune-related genes and disease resistance in juvenile Nile tilapia, Oreochromis niloticus. Aquacult. Nutr. 2019, 25, (3), 597-608.
Reviewer 2 Report
Comments and Suggestions for Authors
: Ultimately, even when vitamin E was added to vegetable oil, it was significantly lower in growth than fish oil. Vitamin E supplementation studies in tilapia are not new. Studies on reproduction, growth, immunity, and stress following vitamin E deficiency have also been addressed in previous studies. Replacing fish oil with olive oil has also been done in several studies. As an example, Zhang et al. (2021), the study on Vitamin E is similar to the purpose of this study. This study was not referred to in this study, but when compared to this study, it seems necessary to explain what is different in proving the effect of vitamin E on tilapia. It is considered insufficient to refer to existing studies in the discussion.
Zhang, X., Ma, Y., Xiao, J., Zhong, H., Guo, Z., Zhou, C., ... & Liu, T. (2021). Effects of vitamin E on the reproductive performance of female and male Nile tilapia (Oreochromis niloticus) at the physiological and molecular levels. Aquaculture Research, 52(8), 3518-3531.
I send a message of support for the efforts of the authors.
Materials and methods
Page 3, Line 108
: In the main text, I read that the feed manufacturing process was referred to in previous study, but I don't think the exact feed manufacturing equipment and method are listed in previous study. In the case of a vitamin-added feed experiment, it is a very important part how vitamins were added to the feed, so it should be described in detail.
Page 3, Line 110, Table 1
: If you add up all the feed composition tables, it is over 1,000, and it comes out as 1,005. There is no numerical value for coconut oil. Why did you include coconut oil in the feed composition table? Since it is a feed addition experiment, the composition table must be accurate.

Author Response
Dear editors and reviewers,
Thank you for your letter and comments on our manuscript titled "Vitamin E ameliorates impaired ovarian development, oxidative stress, and disrupted lipid metabolism in Oreochromis niloticus fed with a diet containing olive oil instead of fish oil" (antioxidants-2489865). These comments helped us improve our manuscript and provided important guidance for future research.
We have addressed the editor’s and reviewers’ comments to the best of our abilities. We hope our manuscript now meets your requirements for publication.
We marked the revised portions in red in our manuscript. All comments and our specific responses are detailed below.
Reviewer #2: Comments on antioxidants-2489865:
Comment 1: Ultimately, even when vitamin E was added to vegetable oil, it was significantly lower in growth than fish oil. Vitamin E supplementation studies in tilapia are not new. Studies on reproduction, growth, immunity, and stress following vitamin E deficiency have also been addressed in previous studies. Replacing fish oil with olive oil has also been done in several studies. As an example, Zhang et al. (2021), the study on Vitamin E is similar to the purpose of this study. This study was not referred to in this study, but when compared to this study, it seems necessary to explain what is different in proving the effect of vitamin E on tilapia. It is considered insufficient to refer to existing studies in the discussion.
Zhang, X., Ma, Y., Xiao, J., Zhong, H., Guo, Z., Zhou, C., ... & Liu, T. (2021). Effects of vitamin E on the reproductive performance of female and male Nile tilapia (Oreochromis niloticus) at the physiological and molecular levels. Aquaculture Research, 52(8), 3518-3531.
Response 1: We revised and referred to more existing studies in introduction and discussion part. The revised part is in page [2], lines [86–88], lines [90–94]; page [13], lines [365–367]; page [14], lines [400–406]; page [15], lines [441–445]; page [16], line [503], lines [510–512];
Comment 2: In the main text, I read that the feed manufacturing process was referred to in previous study, but I don't think the exact feed manufacturing equipment and method are listed in previous study. In the case of a vitamin-added feed experiment, it is a very important part how vitamins were added to the feed, so it should be described in detail.
Response 2: We revised and added more detailed information in section 2.2 Experimental diet preparation. The revised part is in page [3], lines [113–119].
Comment 3: If you add up all the feed composition tables, it is over 1,000, and it comes out as 1,005. There is no numerical value for coconut oil. Why did you include coconut oil in the feed composition table? Since it is a feed addition experiment, the composition table must be accurate.
Response 3: We apologize for the confusion caused by the mention of coconut oil in the previous context. It has been duly noted that coconut oil is not relevant to the manuscript and has been removed. Upon reevaluation of our experimental feed formula (Table 1), we have identified a mistake. It has come to our attention that during the transcription of paper data, there was an error in recording the number '2'. Instead, the number '7' was mistakenly used in the components of α-cellulose. we have rectified this mistake and made the necessary corrections. The revised part is in page [3], lines [120];
Round 2
Reviewer 1 Report
These revisions are fine. However, the authors need to change RO to FO in a few places where revisions were added.
Author Response
Reviewer #1: Comments on antioxidants-2489865:
Comment 1: These revisions are fine. However, the authors need to change RO to FO in a few places where revisions were added.
Response 1: We found the referee’s comment most helpful and have revised the manuscript. The revised part is in page [2], lines [86-89].